# Extracellular Vesicles from Uterine Aspirates Represent a Promising Source for Screening Markers of Gynecologic Cancers

**DOI:** 10.3390/cells11071064

**Published:** 2022-03-22

**Authors:** Gleb O. Skryabin, Andrey V. Komelkov, Kirill I. Zhordania, Dmitry V. Bagrov, Svetlana V. Vinokurova, Sergey A. Galetsky, Nadezhda V. Elkina, Darya A. Denisova, Adel D. Enikeev, Elena M. Tchevkina

**Affiliations:** 1Institute of Carcinogenesis, N.N. Blokhin National Medical Research Center of Oncology, Kashirskoye sh. 24, 115478 Moscow, Russia; goskryabin@gmail.com (G.O.S.); kiazo2@yandex.ru (K.I.Z.); vinokourova@mail.ru (S.V.V.); sg8126@mail.ru (S.A.G.); muuu222-222@mail.ru (N.V.E.); darhance@gmail.com (D.A.D.); adelbufyeni@mail.ru (A.D.E.); tchevkina@mail.ru (E.M.T.); 2Department of Bioengineering, Faculty of Biology, M.V. Lomonosov Moscow State University, Leninskie Gory, 1/12, 111234 Moscow, Russia; bagrov@mail.bio.msu.ru

**Keywords:** miRNA, ovarian cancer, uterine aspirates, extracellular vesicles, exosomes, small RNA sequencing

## Abstract

Extracellular vesicles (EVs), including exosomes, are key factors of intercellular communication, performing both local and distant transfers of bioactive molecules. The increasingly obvious role of EVs in carcinogenesis, similarity of molecular signatures with parental cells, precise selection and high stability of cargo molecules make exosomes a promising source of liquid biopsy markers for cancer diagnosis. The uterine cavity fluid, unlike blood, urine and other body fluids commonly used to study EVs, is of local origin and therefore enriched in EVs secreted by cells of the female reproductive tract. Here, we show that EVs, including those corresponding to exosomes, could be isolated from individual samples of uterine aspirates (UA) obtained from epithelial ovarian cancer (EOC) patients and healthy donors using the ultracentrifugation technique. First, the conducted profiling of small RNAs (small RNA-seq) from UA-derived EVs demonstrated the presence of non-coding RNA molecules belonging to various classes. The analysis of the miRNA content in EVs from UA performed on a pilot sample revealed significant differences in the expression levels of a number of miRNAs in EVs obtained from EOC patients compared to healthy individuals. The results open up prospects for using UA-derived EVs as a source of markers for the diagnostics of gynecological cancers, including EOC.

## 1. Introduction

Extracellular vesicles (EVs), and especially exosomes, are of growing interest in the context of the search for markers for diagnosing and staging malignant tumors, assessing disease prognosis and predicting response to therapy. Exosomes are nanosized vesicles that differ from other EV types in the way of biogenesis; they originate in the vesicular traffic system and are released into the extracellular space upon the fusion of multivesicular endosome membranes with the plasma membrane [1]. Small EVs corresponding to exosomes are found in all biological fluids and contain various classes of biomolecules, including key signaling proteins and various types of RNA (including coding and regulatory RNAs). Exosomal markers of cancer have a number of advantages over both tissue and serological markers [2,3]. One of the most commonly used sources of small EVs for the search of cancer markers is blood plasma, but it contains an extremely heterogeneous pool of vesicles secreted by numerous blood cells, immune cells, epithelial and other cells, from which exosomes that originated from cancer cells consist of a very small percentage [4,5]. In contrast, EVs from “local” body fluids such as ascites and pleural fluid, arising in the areas of primary tumor growth, are enriched in tumor-originated exosomes, making them a more suitable source of cancer markers. However, there is no sufficient normal control for such EV sources (i.e., the same biological fluids of healthy individuals), which strongly limits an adequate comparison when looking for cancer markers. In this respect, uterine aspirates (UA) appear to be a very promising source of EVs for the study of gynecological cancer markers. To our surprise, we found no previously published results on this topic. Earlier, in 2016, the feasibility of isolating exosome-like vesicles from this biofluid was demonstrated for the first time [6]. However, for further RNA isolation and RT-qPCR analysis, the authors used pooled UA specimens obtained from different individuals. That was probably due to the small volume of individual UA specimens, making it difficult to isolate EVs in sufficient quantity for a subsequent analysis. In a few later published studies unrelated to cancer research, uterine flushes have been used as a similar source of exosomes [7,8].

We show for the first time that EVs, including small EVs corresponding to exosomes, can be isolated from individual UA samples from both cancer patients and healthy donors using the ultracentrifugation method with some slight modifications. Interestingly, their concentrations were significantly higher than that isolated under the same conditions from uterine flushes (UF). The vesicles obtained from UA correspond to exosome-like vesicles according to the criteria recommended by ISEV (International Society for Extracellular Vesicles) [9], including size, morphology (based on TEM and NTA data) and enrichment in exosomal markers belonging to distinct functional groups of proteins (according to immunoblotting data). The number of vesicles isolated from UA was sufficient not only for protein isolation and analysis but also for obtaining RNA preparations, containing different types of RNA, including noncoding regulatory RNAs of various classes, which was confirmed by the analysis of RNA size distribution and sequencing data. NGS-based small RNA-seq revealed the presence of a wide range of RNA species, including miRNAs and a number of other regulatory molecules. The comparison of miRNA sequencing data obtained using a small pilot sample revealed the panel of differently expressed miRNAs in EVs from healthy donors and patients with epithelial ovarian cancer (EOC, a serous adenocarcinoma subtype). The choice of ovarian cancer (OC) is based both on the extremely high demand for markers of a liquid biopsy for early diagnosis and on several lines of evidence linking changes in the cellular and molecular compositions of the uterine cavity fluid into the pathogenesis of ovarian cancer [10,11,12,13].

Differences in the levels of a few selected miRNAs (miR451a, miR199a-3p and miR375-3p) in EVs from EOC patients and healthy donors were further confirmed by stem-loop RT-qPCR. Overall, these results show that UA represent a promising source for studying exosome-like EVs. Given that the sampling procedure is safe and noninvasive, this biofluid can be used for screening for exosomal markers of gynecological cancers.

## 2. Materials and Methods

### 2.1. Clinical Specimens and Patient Consent

Uterine aspirates and flushes from the uterine cavity were collected using a type C Pipelle probe from epithelial ovarian cancer patients (EOC patients, N = 5) and patients with no history of cancer (control group, N = 5). Samples were received from the Gynecologic Department of the N.N. Blokhin National Medical Research Center of Oncology. The EOC samples included high-grade serous adenocarcinoma (TNM status and disease stages for each patient are shown in Table 1). Clinical diagnosis in each case was confirmed by histopathology. The tumor clinic morphological stages were determined according to the FIGO classification. Sampling was performed before the surgery or other treatment. Written informed consent was sought and obtained from all participants in accordance with the N.N. Blokhin National Medical Research Center of Oncology Ethics Committee guidelines.

The initial volume of UA ranged from 200 µL to 1.5 mL. The obtained samples were diluted in 5 mL of ice-cold PBS right after collection. Alternatively, uterine flushes were obtained using a technique in which a sterile saline solution was injected using dual-channel catheters, and 8-mL flushes were taken.

### 2.2. Sample Processing

All the clinical specimens were processed no more than two hours after the sampling. Samples were placed in 15mL centrifuge tubes and kept on ice through all the processing. After 30 s of vortexing, samples were centrifuged consecutively at 300× *g* (4 °C) for 15 min, 800× *g* (4 °C) for 15 min, 2000× *g* (4 °C) for 20 min A-4-81 rotor (Eppendorf Centrifuge 5810R) and 10,000× *g* (4 °C) for 30 min F-34-6-38 rotor (Eppendorf Centrifuge 5810R) to remove cells, debris and large particles. The supernatants obtained in the last step were frozen at −80 °C until needed.

### 2.3. Isolation of EVs

We used a standard protocol of differential centrifugation described by Thery et al. [14] with slight modifications. Briefly, thawed supernatants from UA and UF samples were diluted with ice-cold PBS or saline, respectively, to a final volume of 35 mL and transferred to ultracentrifuge tubes (#326823, Beckman Coulter, Brea, CA, USA) to perform the first ultracentrifugation round at 110,000× *g* (4 °C) for 2 h with a SW-28 swinging bucket rotor (*k* factor 245.5; Beckman Coulter). The obtained pellet (containing mostly small EVs) was resuspended in 5 mL of ice-cold PBS and transferred to small ultracentrifuge tubes (#326819, Beckman Coulter) and centrifuged again at 110,000× *g* (4 °C) for 1 h with a SW-50.1 swinging bucket rotor (*k* factor 154.5; Beckman Coulter). The final cleared pellet was resuspended in 120 μL of ice-cold PBS (#70011-044, Gibco, Grand Island, NY, USA); aliquoted in Eppendorf tubes (Protein LoBind #022431005) for nanoparticle tracking analysis (NTA), transmission electron microscopy (TEM), RNA isolation and protein analysis; frozen in liquid nitrogen and stored at −80 °C for further analysis.

### 2.4. Particle Size Distribution and Quantification

The size distribution and concentration of EVs were determined by NTA using a NanoSight LM10 HS instrument equipped with a NanoSight LM14 unit with onboard temperature control (Malvern Panalytical Ltd., Malvern, UK), a LM 14C (405 nm, 65 mW) laser unit and a high-sensitivity camera with a scientific CMOS sensor (C11440-50B, Hamamatsu Photonics, Hamamatsu City, Japan). All measurements were performed in accordance with ASTM E2834–12(2018), with the following camera and video processing setups optimized for EV measurement: Camera Shutter = 1500, Camera Gain = 500, Lower Threshold = 195, Higher Threshold = 1885, Screen Gain = 10 and Detection Threshold = 8 (Multi). Each sample was diluted with particle-free PBS down to a concentration of about 1.5 × 10^8^ particles/mL. Twelve videos 60 s long each were recorded and processed using NTA software 2.3 build 33 (Malvern Panalytical Ltd.). The results from all measurements were combined to obtain a particle size histogram and the total particle concentration corrected for the dilution factor using the NTA software feature.

### 2.5. Transmission Electron Microscopy

The carbon-coated TEM grids (Ted Pella) were treated for 45 s using an Emitech K100X glow discharge device (Quorum Technologies Ltd., Laughton, UK) to make the carbon surface hydrophilic and increase the adsorption of the vesicles. The samples of isolated vesicles were 5- to 40-fold diluted (depending on the NTA-measured concentration) in PBS to ensure a convenient surface density of the adsorbed particles. The samples were deposited for 30–60 s, stained with 1% uranyl acetate for 45 s twice and dried. Images (at least 10 per sample) were obtained using a JEM-1011 transmission electron microscope (JEOL, Ltd., Akishima, Japan) operating at 80 kV.

### 2.6. Immunoblotting and Antibodies

The total protein concentration in EV samples and cells lysed in RIPA buffer were determined using Bradford reagent (#500-0006, Bio-Rad Laboratories, GmbH, Munich, Germany), according to the manufacturer’s recommendations using a Benchmark Plus microplate spectrophotometer (Bio-Rad Laboratories). Western blotting was performed according to the previously described procedure [15]. Briefly, ten micrograms of total protein were applied to 15% SDS-PAGE, transferred to a PVDF membrane (Merk Millipore Ltd., Carrigtwohill, Ireland) and blocked with 5% BSA (#0332-100G, VWR Life Science, Solon, OH, USA) in TBS buffer with 0.1% TWEEN-20 (MP Biomedicals, LLC, Irvine, CA, USA) for 1 h at room temperature (RT). Membranes were incubated overnight at 4 °C with primary antibodies, washed three times in TBS/TWEEN-20 and incubated with secondary antibodies (1 h, RT). Proteins were visualized with an Enhanced Chemiluminescence Detection Kit (Millipore). Images were captured using a Kodak GelLogic 2200 Imaging System. The following primary and secondary antibodies and dilutions were used: anti-Alix (#sc-271975, 1:500; Santa Cruz Biotechnology, Dallas, TX, USA), anti-Flotillin-2 (#3436S, 1:1000; Cell Signaling Technology, Topsfield, MA, USA), anti-CD9 (#13174, 1:2000; Cell Signaling Technology), anti-TSG-101 (ab125011, 1:5000; Abcam, Cambridge, UK), anti-Caveolin-1 (ab2910, 1:1000; Abcam), anti-PCNA (#sc-7907, 1:500; Santa Cruz Biotechnology), anti-mouse goat polyclonal antibodies (#2367, 1:5000; Cell Signaling Technology) and anti-rabbit goat polyclonal antibodies (#29902, 1:80,000; Cell Signaling Technology).

### 2.7. RNA Extraction and Measurement

The RNA from EVs was isolated using a Total Exosome RNA & Protein Isolation Kit (#4478545; Invitrogen, Vilnius, Lithuania), according to the manufacturer’s protocols. RNA was eluted from the last column with 60 µL of nuclease-free water and stored at −80 °C until further analysis. A small RNA concentration was measured using the Qubit™ microRNA Assay Kit (Q32881; Invitrogen) according to the manufacturer’s protocol. The amount, quality and size distribution of isolated RNA were analyzed by an Agilent 2100 Bioanalyzer for small RNA profiles using Small RNA Kits (Agilent Technologies, Santa Clara, CA, USA).

### 2.8. Small RNA Deep Sequencing

Total RNA, containing the small RNA fraction, has been converted into cDNA libraries using 2 µL of total exosomal RNA according to the NEBNext^®^ Multiplex Small RNA Library Prep Set for Illumina (NEB) (BioLabs Inc., Hitchin, UK) for all the preparations. Next, we measured the DNA yield and distribution of the fragment lengths of the obtained libraries using the High Sensitivity DNA Kit on a Bioanalyzer 2100 (Agilent Technologies) and carried out the selection cDNA according to the miRNA size using the AMPure XP Beads (BioLabs Inc.) according to the NEB. Sequencing was performed on a HiSeq1500 (Illumina) instrument by single-end reads of 50 nucleotide lengths with the generation of at least 5 million reads mapped to the human genome (human genome (hg18) assembly version).

### 2.9. Bioinformatics

Analysis of the obtained sequence results in FASTA files was performed as follows: the pipeline was designed to preprocess raw reads to trim and remove adapter sequences and generate high-quality reads that were mapped to the human genome assembly hg18 (https://www.ncbi.nlm.nih.gov/assembly/GCF_000001405.12/, accessed on 27 December 2021), miRBase v22.1 (https://www.mirbase.org/, accessed on 5 January 2022) and piRBase v2 (http://bigdata.ibp.ac.cn/piRBase/, accessed 12 January 2022). Aligned reads were classified according Ensembl data annotations (https://www.ensembl.org/info/genome/genebuild/biotypes.html, accessed on 22 January 2022). Adapters were removed using cutadapt software (v1.12) (parameters—trim-n-n5-m14), mapped to the genome assemble hg18 with bowtie software (v0.12.9) (mapping parameters-n 0) and counted the number of reads with featureCounts (v1.5.0) (parameters-g transcript_id-t miRNA-Q 10-a hsa.mature.gtf) using the annotations mentioned above. We next used specialized software package edgeR (version Galaxy 3.34.0) to analyze the differential expression of miRNAs [16].

### 2.10. Reverse Transcription and Quantitative Real-Time PCR

Stem-loop RT-PCR for miRNA quantification was performed according to the method described by Chen et al. [17]. All UA-exo miRNA samples were diluted to equal concentrations, and 4 ng of exosomal miRNA was reverse-transcribed using a TaqMan™ MicroRNA Reverse Transcription Kit (#4366596, Applied Biosystems, Vilnius, Lithuania) with 1 pmol of miRNA-specific stem-loop RT primers. A minus reverse transcriptase control was performed for each sample to assess the DNA contamination. Reverse transcription products were two-times diluted with nuclease-free water prior to the RT-qPCR performance. Primers for specific miRNAs were designed using miRBase v22.1 and synthesized by “DNA synthesis” (Moscow, Russia) (sequences are shown in Appendix A Appendix A). The amplification efficiencies were tested using serial dilutions of cDNA, obtained from the reverse transcription reaction of corresponding synthesized miRNAs. The PCR parameters were set to accomplish an efficiency between 95% and 105% for all primer sets. To validate the RNA-sequencing data, we conducted a qPCR analysis of miR-199a-3p, miR-451a, miR-375-3p, let-7b-5p, miR-16-5p and miR-23a-3p.

RT-qPCR was performed by CFX96 Real-Time PCR Systems (Bio-Rad) using TaqMan™ Universal MasterMix II (#4440038, Applied Biosystems) with 20 pmol of forward primers, 10 pmol of universal reverse primer and 6 pmol of TaqMan™ miRNA-specific probes in a 20-µL reaction. The cycling conditions were as follows: 50 °C for 2 min, 95 °C for 10 min, followed by 50 cycles of 95 °C for 15 s and Tm for 1 min. The melting temperatures (Tm) were 59 °C for miR-199a, miR-16 and let-7b; 58 °C for miR-375; 54 °C for miR-451a and 60 °C for miR-23a. The reactions were performed in triplicate, and results with a standard deviation value < 0.37 were accepted. Bio-Rad CFX Manager software v.3.1 was used for data analysis and threshold cycle (Ct) value calculations. miRNA expression data were normalized to miR-23a. Fold changes (FC) were determined using the ΔΔCt method, where ΔCt = Ct(miRNA) − Ct(miR-23a) and ΔΔC(t) = ΔCt(sample) − average ΔCt(control) and FC = 2^−ΔΔCt^.

### 2.11. Statistical Analysis

A modified version of Fisher’s exact test was used to evaluate the statistical significance of differential miRNA expression [16]. Based on NTA-measured particle size and concentration, values of the mean, mode, percentile data (10th and 90th), standard deviation and confidence interval were calculated using Wolfram Mathematica software. The Student’s *t*-test and analysis of variance (ANOVA) were used for the comparison of groups. The Benjamini–Hochberg method was used to control the expected proportion of false hypothesis rejections (FDR), and the obtained value was used to assess the significance of the result. *p*-values lower than 0.05 were considered statistically significant. For statistical analysis, we used specialized software package edgeR (version Galaxy 3.34.0), statistical software package GraphPad Prism ver. 8.0.0 package for Windows and the engineering–mathematical package Wolfram Mathematica ver. 11. The package MS Excel 2016 for Windows was used for plotting graphs.

## 3. Results

### 3.1. Characterization of EVs from Uterine Aspirates

EV preparations were obtained from uterine aspirates and uterine flushes collected from EOC patients and individuals without a history of cancer. EVs were isolated using the ultracentrifugation-based method. The size and concentration of the EVs were analyzed by Nanoparticles Tracking Analysis (Figure 1A,B). EVs’ size and morphology were studied by transmission electron microscopy (Figure 1C).

Images of nanoparticles corresponding to exosomes (“cup-shaped” morphology typical for exosome sizes) were obtained with TEM for all preparations. Surprisingly, the total amount of EVs extracted from UF was essentially lower than that from UA in all cases (see Figure 1A,C). We did not conduct the precise comparison of concentration of EVs isolated from UA and UF, as both fluids cannot be collected from the same patient, and such a task would therefore require an additional study using two independent samplings. However, our observation is that the UA contain higher amounts of EVs, probably due to the fact that most of the EVs are contained in the mucus component and are not washed out by the saline. Therefore, for further research, we used UA specimens (N = 10), including five specimens obtained from patients with a high-grade serous subtype of epithelial ovarian cancer (the clinical characteristics are summarized in Table 1) and five specimens from individuals without a history of cancer (healthy individuals).

The mean size of the particles according to the NTA data varied from 97 to 135 nm, with a mode from 67 to 115 nm in different individual EV preparations. The mean size of the EVs and median over the entire sampling of the preparations were 129.5 nm (SD 7.0) and 113.5 (SD 8.4), respectively (Figure 1B). The EV concentration in the preparations varied from 10^11^ to 10^13^ particles per mL (the average EV concentration in UA was 5.60 × 10^12^ particles/mL). The mean size of the EVs, as well as their concentration in preparations obtained from EOC patients and healthy individuals, had no significant differences (Student’s *t*-test, *p* > 0.05).

To confirm the exosomal nature of the vesicles, we further analyzed exosomal markers in EV preparations by Western blotting. For this task, proteins belonging to different functional classes and with different intracellular compartmentalizations were selected in accordance with ISEV recommendations [9]. The results of the analysis of exosomal markers, including TSG-101 and Alix (components of different subunits of the ESCRT complexes) and tetraspanin CD9, as well as Flotillin-2 and Caveolin-1 (components of flat lipid rafts and caveolae), are shown in Figure 2. The PCNA protein was chosen as a negative control (control for the presence of non-exosomal proteins in EV preparations). The lysate of the ovarian cancer cells EFO-21 was used for the comparison of protein levels in the EVs and cells. All proteins were analyzed in a single experiment, making it possible to compare the ratio of the studied exosomal markers in different EV preparations.

The results showed that the EVs are enriched in all exosomal markers, except for Alix, the level of which varied considerably in different EV preparations and was undetectable in some cases. No differences were found in the content of the exosomal markers between EVs obtained from healthy subjects and EOC patients.

### 3.2. Characterization of Small RNAs Obtained from EVs Isolated from Uterine Aspirates

The obtained and characterized EV samples were further used to analyze the composition of small RNAs. RNA was isolated using the Total Exosome RNA & Protein Isolation Kit (#4478545; Invitrogen). The small RNA content and size distribution were analyzed by capillary microelectrophoresis using the Small RNA kit for 2100 Bioanalyzer Instrument (Agilent Technologies). The results showed a wide size spectrum of small RNAs, including a peak of about 23 nucleotides corresponding to miRNAs (Figure 3A). The ratio of RNA of different sizes varied significantly between EV samples.

Next, we analyzed the composition of small RNA in EVs using NGS small RNA-seq. The results of the analysis using the corresponding databases (described in Section 2.9 Bioinformatics) confirmed the presence of various RNA species, including regulatory RNAs, such as snRNA, snoRNA, piRNA, vault RNA, siRNA and scRNA, as well as short fragments originating from protein coding and structural RNAs, such as rRNA, mRNA, lncRNAs, various pseudogenes and intergenic repeats, etc. The ratio of different RNA classes varied markedly in different EV preparations. The mean share percentage of different RNA types is shown in Figure 3B. The proportion of miRNA in the small RNA transcriptome ranged from 1.10% to 7.50% in different individual EV samples, averaging 4.95%.

### 3.3. Comparison of EV miRNAs Content in UA from EOC Patients and Healthy Individuals

Profiling of the miRNAs revealed more than 700 individual miRNAs presented in EVs from the UA. The most abundant miRNAs are shown in Figure 4A. Comparison of the miRNA content in EVs from cancer patients and healthy individuals revealed 57 differentially expressed miRNAs (FDR-adjusted *p* < 0.05), including 30 downregulated and 26 upregulated. The MDS plot (multidimensional scaling plot) shows variations among the miRNA difference expressions shown in Figure 4B.

Using additional criteria beyond validity (FDR-adjusted *p* < 0.05), such as a high degree of difference (fold change > four) and high level of representation in EVs (logCPM > 9), we selected 29 differentially expressed miRNAs, including 18 upregulated and 11 downregulated miRNAs in EVs of EOC patients compared to healthy donors (Table 2). The list of upregulated miRNAs in a sequence of absolute fold change includes: miR-1246, miR-200b-5p, miR-375-3p, miR-320a-3p, miR-183-5p, miR-320b, miR-200c-3p, miR-224-5p, miR-125a-5p, miR-320c, miR-182-5p, miR-200b-3p, miR-9-5p, miR-125b-5p, let-7b-5p, miR-429, miR-10a-5p and miR-141-3p. The list of downregulated miRNAs in the sequence of absolute fold change includes: miR-451a, miR-542-3p, miR-424-5p, miR-450b-5p, miR-449c-5p, miR-411-5p, miR-19b-3p, miR-196b-5p, miR-143-3p, miR-23b-3p and miR-199a-3p. Notably, among the differently expressed miRNAs were known cancer-associated molecules, such as members of the miR-200 family, including miR-200c, miR-200b, miR-141 and miR-429, and some others.

To verify the NGS data, we next performed a selective comparison of the levels of several individual miRNAs from among those differentially expressed in EVs by using stem-loop RT-qPCR.

For this task, we chose miRNAs from among the most and least differently expressed in the comparison groups, according to the NGS data. miR451a and miR199a-3p were chosen from the list of downregulated and miR375-3p from the upregulated molecules.

Based on the NGS results, we did not find known reference molecules to normalize the RT-qPCR data, i.e., those that would fully meet the criteria for miRNA normalization. For example, miR16-5p and let-7b-5p, often used for such purpose, were expressed differently in different EV samples according to both small-RNA-seq and RT-qPCR (Appendix A Appendix A). One of the most equally expressed miRNAs according to the sequencing and RT-qPCR data was miR23a-3p, which we chose as the reference for the assessment of miR451a, miR199a-3p and miR375-3p expression using the ΔCt parameter. The differences between the studied miRNA levels in the compared groups (fold change) was calculated using the ΔΔCt method. As shown in Figure 4C, miR451a and miR199a were 2.94- and 7.89-fold decreased, respectively, while miR375-3p was 3.05-fold increased in EVs from EOC patients compared to the control group (*p* < 0.05). Thus, the RT-qPCR data are consistent with the results of deep sequencing and confirmed significant differences in the miRNA expression profiles in EVs from uterine aspirates from EOC patients and healthy individuals.

## 4. Discussion

There are several reasons for the great interest in EVs, especially exosomes, in terms of cancer markers. First, molecular markers within EVs appear to be more informative than tissue markers with regards to the search for cancer markers [18]. Since a tumor is a heterogeneous population containing cells at different stages of EMT and progression in general, the molecular composition of biopsy specimens may be fragmentary, i.e., correspond only to a part of the tumor cell population. At the same time, EVs found in body fluids are produced by all tumor cells, as well as by tumor-associated stromal cells and can thus more fully reflect tumor-associated disturbances, including epigenetic changes and signaling pathway alterations. Compared with liquid cancer biopsy markers—in particular, cell-free nucleic acids—EV molecules also have several advantages, such as: (i) high stability due to the surrounding bilayer lipid membrane [19], (ii) enrichment with molecules, particularly RNA and miRNA, compared to non-vesicular fractions of the same body fluids [20,21] and even compared to parental cells [22] and (iii) high specificity, since the cell-free molecules detected in circulation may reflect cell damage or death, whereas EV cargo molecules are the result of precise selection and targeted loading [23,24].

Regarding RNA, numerous data indicate that cells destine specific molecules for extracellular release within EVs. For instance, certain miRNAs have been shown to be preferentially exported and enriched in EVs [25]. This is also confirmed by differences in the composition of miRNAs between different populations of EVs produced by the same cells [26,27], as well as by differences between the miRNA profiles of cells and the vesicles they secrete [22,25,28]. For miRNAs, there are four putative mechanisms for loading into EVs, including the ceramide-dependent pathway, also known as the neutral sphingomyelinase 2 (nSMase2)-dependent pathway [29,30,31], the ribonucleoprotein (hnRNP)-dependent pathway requiring the involvement of EXO motifs in miRNAs and the sumoylation of heterogeneous nuclear ribonucleoprotein A2B1 (hnRNPA2B1) [32,33], the 3′-end miRNA sequence-dependent mechanism [34,35] and the miRNA-induced silencing complex (miRISC)-related pathway [36,37,38].

Since the discovery that exosomes are involved in the transport of extracellular RNAs [39,40], there has been ample evidence that exosomal miRNAs may be a very promising source of disease biomarkers, including cancer markers [41]. At the same time, progress in this direction is rather slow, which can be explained by a number of reasons. One of them is the extremely high natural heterogeneity of EVs, including various types of vesicles (microvesicles, exosomes, apoptotic corpuscles, etc.), which cannot always be clearly distinguished by size, density, morphology and other physical characteristics [21,42]. Furthermore, a great diversity in molecular composition is shown even for EVs with similar physical characteristics and originating from the same cells [43,44,45].

Another reason for the heterogeneity of data on the molecular composition of EVs in general and their miRNA content in particular is the wide variety of techniques used in different studies for isolating and analyzing both the EVs themselves and their molecular compositions [46]. Such a variety of methodological approaches causes a great deal of inconsistency in the results and severely limits the research on EVs in general. In the context of EV cancer markers, another major problem is the lack of hitherto generally accepted criteria and methods for isolating EVs of tumor origin. This problem is further complicated by the fact that the vast majority of studies use blood (plasma or serum), less often urine, as the source of EVs. An obvious advantage of such body fluids is the simplicity and safety of sampling, as well as the relatively large sample volume. However, blood as a source of EVs has a number of limitations. First of all, the major producers of EVs in the circulation are blood cells and immune cells, as well as vascular endothelial cells and a number of other epithelial cells [19]. Only a very small proportion of vesicles in the total pool of EVs in the blood is of tumor origin (probably less than 1%) [4,5]. In addition, a large number of ribonucleoprotein complexes and lipoproteins (including HDL and LDL) with similar physical characteristics to exosomes are present in blood, which almost inevitably contaminate the EV preparations with proteins and RNAs of non-vesicular origin [42].

In view of the above, more promising sources of EVs for the search of cancer markers seem to be “local” fluids formed in the areas of primary tumor growth, because the percentage of EVs of tumor origin in such fluids should be significantly higher. These body fluids include ascitic, pleural, peritoneal and some other fluids [47,48,49,50]. However, they are formed due to pathological processes (such as cancer and inflammatory diseases) and have no counterpart in healthy individuals. For this reason, they appear to be a useful source of EV markers for tumor staging, prognosis assessment and prediction of the response to therapy but are unlikely to be a suitable source of diagnostic markers, especially for early cancer detection.

Uterine aspirates appear to be a promising source of EVs for the search for diagnostic markers of gynecological cancers because, on the one hand, they should contain a high proportion of EVs produced by epithelial cells of the female reproductive tract and, on the other hand, can be obtained from both cancer patients and healthy individuals. To our surprise, to date, there have been only a handful of papers analyzing the composition of EVs from human uterine fluid, all of which have not been related to cancer research. In 2016, Campoy et al. first showed the principle possibility of isolating exosomes from uterine aspirates [6]. However, for RNA isolation from EVs and subsequent analysis by RT-qPCR, the authors pooled preparations from different individuals, which was probably due to the small volume of individual samples and insufficient amount of EVs for RNA isolation and further analysis. In an earlier study, the presence of EVs in uterine lavages was also shown, however, the vesicles were not fully evaluated, according to the ISEV guidelines, which were defined later [51]. Several later published studies have used flushes from the uterine cavity as a similar source of exosomes [7,8]. Remarkably, when comparing the mucous and the liquid fractions of uterine flushes, as well as the cervical brush-derived material, it was found that the liquid fraction of uterine flushing was enriched in EVs, exosomal proteins and small RNAs [7].

Here, we first conducted the isolation of EVs from uterine aspirates from cancer patients and healthy subjects and compared the miRNA profiles in the obtained EV preparations. The choice of EOC as a type of gynecological cancer was determined due to several reasons. First, ovarian cancer is still the leading cause of cancer deaths in women worldwide, with the majority of cases being diagnosed at stages III to IV, which largely determines the high mortality rate from this disease—the second highest mortality among gynecological cancers worldwide [52,53]. The search for markers for the early diagnosis of OC is therefore an extremely urgent task. The first evidence is emerging that exosomal miRNAs can serve as such markers [54]. Secondly, there is now increasing evidence that clinical material from the uterine cavity can be a source of OC markers. The presence of tubo-ovarian cancer cells in the uterine cavity and cervical canal was first shown as early as 1985 [55]. In later published works, cells of peritoneal carcinoma, fallopian tube cancer and OC of various histological forms were found in uterine flushings and cervical smears [10]. The results of whole-exome sequencing revealed the presence of mutations associated with OC in DNA samples from the Papanicolaou test. The authors found tumor cells in 100% of the clinical samples for endometrial cancer and 41% for OC [13]. Genomic alterations characteristic of OC were also found in lavages of the uterine cavity in 80% of OC patients [12]. In 2018, Wang et al. reported the detection of endometrial and ovarian cancers based on genetic changes in DNA extracted from uterine and cervical fluids [11].

It is also important to emphasize the safety of sampling from the uterine cavity and cervical canal, which allows a noninvasive biopsy to be performed at routine preventive testing, including among women at risk of gynecological cancer. Thus, given the detection of cell morphologically and genetically corresponding to OC in the uterine cavity, along with the increased production of EVs by malignant cells compared to normal cells [56], it can be assumed that uterine aspirates from OC patients will be enriched in EVs originating from tumor cells. In this case, the molecular composition of EVs, including the small RNA content, in the UA of OC patients and healthy subjects will differ, and these differences could be further used to develop OC markers.

To test this hypothesis, we compared miRNA profiles in EVs obtained from UA from EOC patients and healthy donors. The results obtained using a small pilot sample showed significant differences in the expression of more than 50 miRNAs. Using more stringent criteria, including a marked difference in miRNA expression (FC > 4) and high level of representation in the EVs (LogCPM > 9), we revealed 29 differentially expressed RNAs, including 18 upregulated and 11 downregulated in the RNA patient samples.

Interestingly, among the differentially expressed miRNAs, we found a number of known cancer-associated molecules. For example, among the miRNAs elevated in the EOC samples were all members of the miR-200 family, including miR-200b, miR-200c, miR-141, miR-429 and miR-200a (the latter was not included in the list, as the FC value was less than 4 (namely 2.8). miR-200 family members are known tumor promotors, regulating mesenchymal–epithelial transition by targeting E-cadherin [57,58]. The increased expression of miR-200 family in ovarian cancer has been shown repeatedly in both tissues and body fluids of patients [59,60,61,62,63]. Moreover, members of the miR-200 family are considered as promising markers for liquid biopsy of OC, since their increase in blood has been shown in numerous studies [64]. Kan et al. found that miR-200a, miR-200b and miR-200c are highly expressed in sera from serous cancer patients compared to healthy controls, with miR-200c being most significantly different between the two groups [65]. Similarly, the upregulated levels of miR-200a, miR-200b and miR-200c were detected in the sera of EOC patients compared to the healthy group [66]. miR-200c and miR-141 were found significantly elevated in the EOC patient sera when compared to healthy controls [67]. Increased miR-200a and miR-141 levels in the plasma from OC patients compared to the healthy controls has also been confirmed in the study of Zheng et al. [68]. Moreover, the levels of miR-200a, miR-200b and miR-200c in the circulation correlated with the disease stage and reflected their enhanced expression in tissues [69]. Elevated levels of the miR-200 family have also been shown in EVs. Thus, Taylor et al. analyzed miRNA profiles in EpCAM+ exosomes and found increased levels of miR-141, miR-200a, miR-200c, miR-200b, miR-203 and miR-205 in serum from patients with serous papillary adenocarcinoma compared to patients with benign tumors and healthy individuals [70].

Another miRNA that we found to increase dramatically in EVs from EOC patients compared to normal controls is miR-1246. miR-1246 is considered as oncomiR for various cancer types, including OC [71,72]. The diagnostic value of this miRNA has been shown for various cancer types, with elevated levels being detected in the blood, which is of additional interest in the context of liquid biopsy diagnosis [73,74], including the diagnosis of high-grade serous ovarian carcinoma [75]. Increased levels of this miRNA in EVs from body fluids have also been repeatedly shown for various types of cancer [76,77,78].

Among the miRNAs decreased in EOC patients, we found several molecules with known tumor-suppressor activity, such as miR-451a;, miR-199a and some others. Although the role of miR-451a in tumor progression is not well-defined, changes in its expression are characteristic of many types of cancer, most of which show downregulation of this miRNA [79]. Reduced expression of miR-451 has also been shown for EOC. Moreover, low levels of miR-451 were associated with the advanced FIGO stage, high serum CA-125 levels, LN metastasis and poor prognosis for EOC patients [80]. Comparison of the miRNA levels in OC and endometriosis samples revealed significantly reduced levels of miR-451 in ovarian tumors [81]. Ji et al. identified four miRNAs, including miR-451, that could be used to distinguish between OC samples and healthy controls [82]. Notably, several studies on miRNA profiling in tissues and serum of OC patients found a simultaneous increase in the miR-200 family and a decrease in miR-451 compared to the controls [83,84].

Another example of miRNA that has been significantly reduced according to our results in OC-derived EVs is miR-199a. The tumor suppressive activity of this miRNA for OC had been shown previously and appears to be related to the suppression of angiogenesis [85]. Reduced levels of this miRNA in OC cells and tissues have also been shown in other studies, and, as with miR-451a, a decrease in miR-199a was detected simultaneously with an increase in the miR-200 family [86]. Furthermore, the combination of increased expression of miR-200, miR-141, miR-429, miR-18a and miR-93 and decreased expression of miR-199a and let-7b has been associated with poor prognosis in serous ovarian carcinoma [87].

In sum, we showed for the first time that EVs, including small EVs, corresponding to exosomes, can be successfully isolated from individual clinical specimens of uterine aspirates and can be used to search for diagnostic markers of gynecological cancers. Initial experiments on exosomal miRNA profiling, performed by small RNA-seq technique using a pilot set of EOC and control samples revealed a number of differently expressed miRNAs, including miRNA molecules previously shown to be associated with ovarian cancer. Further studies using an expanded sample of uterine aspirates obtained from patients at different stages of the disease will help to identify exosomal miRNAs as potential markers of ovarian cancer, including for early diagnosis of EOC.

## Figures and Tables

**Figure 1 cells-11-01064-f001:**
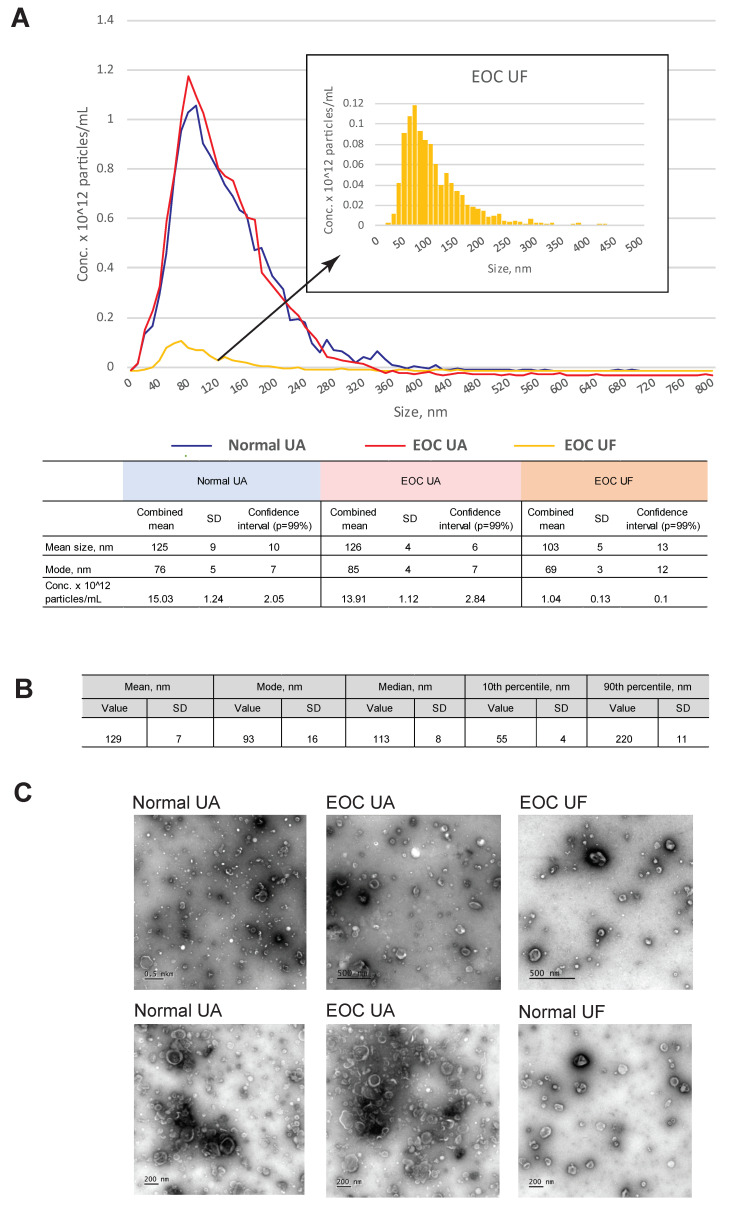
Analysis of the EV size and morphology. (**A**) NTA data for evaluation of the EV size distribution and concentration. Examples of the analysis of EVs isolated from a uterine aspirate of an ovarian cancer patient (EOC UA), a uterine aspirate of a healthy donor (Normal UA) and a uterine flush of an ovarian cancer patient (EOC UF). (**B**) Mean values for the EV size, median and mode over the entire sampling. (**C**) TEM analysis of the EV morphology. Examples of EVs isolated from: uterine aspirates from two healthy donors (Normal UA), uterine aspirate from two EOC patients (EOC UA), uterine flush from an EOC patient (EOC UF) and uterine flush from a healthy donor (Normal UF). The scale bars correspond to 500 nm (upper row) and 200 nm (bottom row).

**Figure 2 cells-11-01064-f002:**
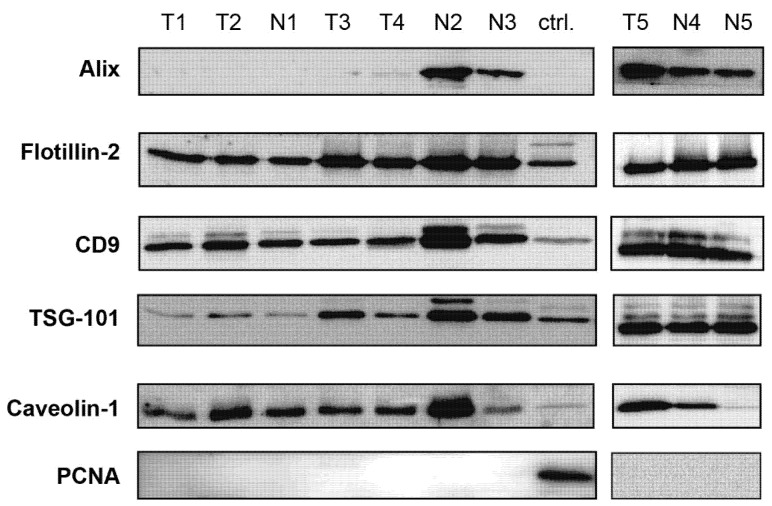
Western blot analysis of exosomal markers TSG-101, Alix, CD9, Flotillin-2 and Caveolin-1 in EVs from the UA of EOC patients (T1–T5) and healthy donors (N1–N5). The PCNA protein was used to confirm the absence of cellular proteins of non-vesicular origin in EV preparations. Protein lysate of EFO-21 cells (ctrl.) was used as a molecular weight control and to compare levels of proteins in cells and EVs.

**Figure 3 cells-11-01064-f003:**
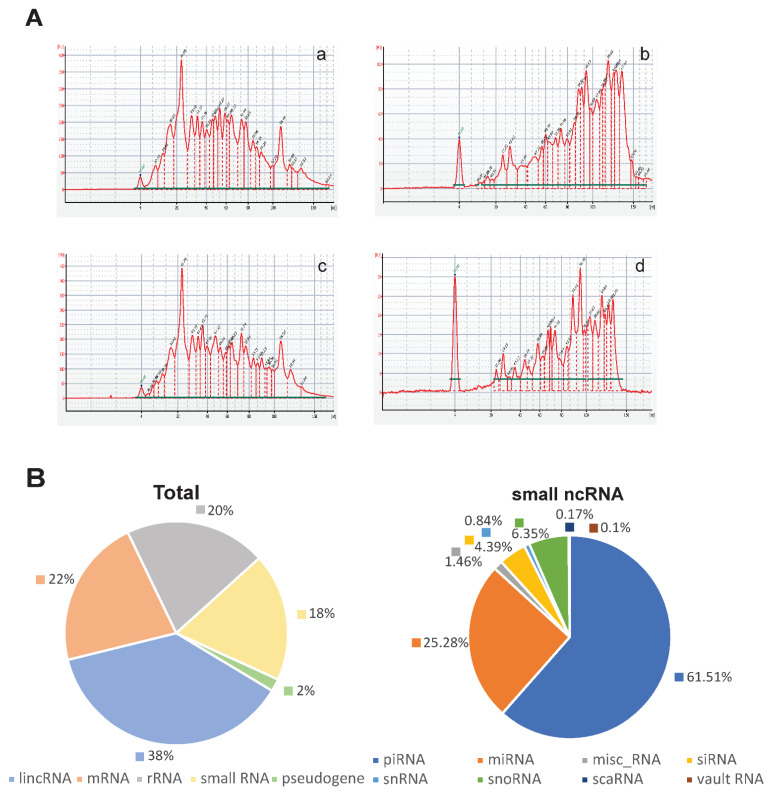
Characteristics of small RNAs present in EVs isolated from UA. Agilent Bioanalyzer profiles of the small RNA. Size distribution and perczentage of the miRNA. (**A**) Examples of small RNA analysis for 4 EV preparations ((**a**,**b**)—from EOC and (**c**,**d**)—from normal UA samples), illustrating relatively high and low contents of molecules corresponding to the sizes of miRNAs. (**B**) RNA content of the EVs according to small RNA-seq data. The mean share percentage for different RNA types is presented.

**Figure 4 cells-11-01064-f004:**
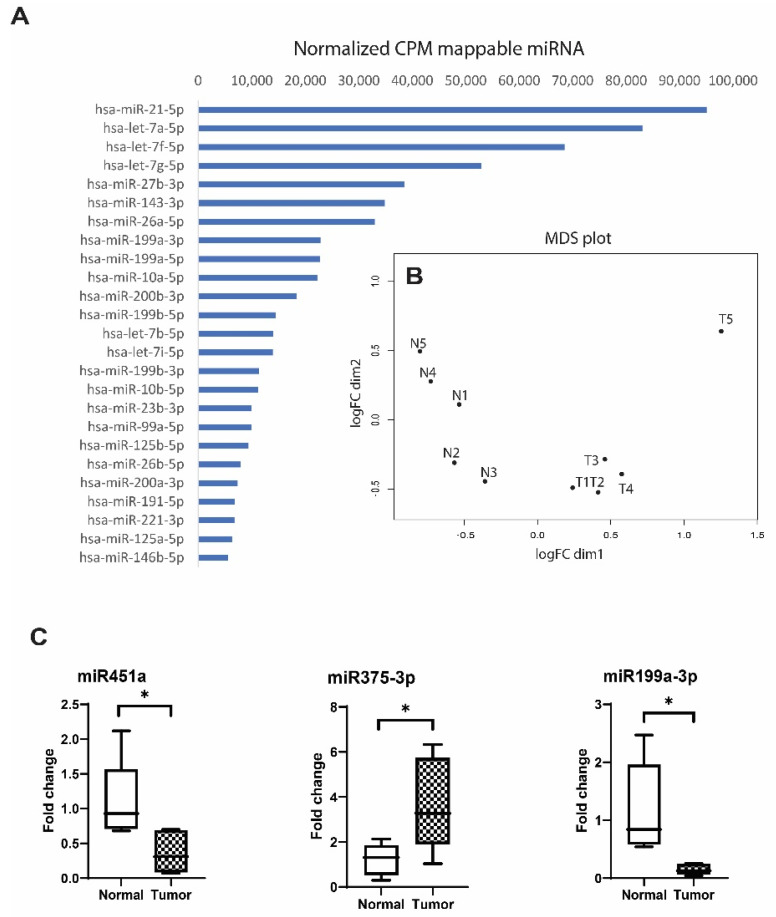
Analysis of miRNAs in EVs isolated from the UA of EOC patients and healthy individuals. (**A**) The 30 most abundant miRNAs among UA exosomal small RNAs (normalized CPM mappable miRNAs). (**B**) MDS plot (multidimensional scaling plot) shows variations among miRNA differential expression in the analyzed samples, and the distance between sample labels indicates a dissimilarity. X- and Y-axes are representative of the Euclidean distances between samples, with the x-axis being dim1 and y-axis being dim2 (logFC); T—EOC samples and N—normal samples. (**C**) Relative expression of miR451a, miR199a-3p and miR375-3p in EVs of EOC patients and healthy individuals from RT-qPCR data. Gene expression data were normalized to miR-23a. Fold changes were calculated as 2^−ΔΔCt^, where ΔCt  = Ct(miRNA) − Ct(miR-23a) and ΔΔC(t) = ΔCt(sample) − average ΔCt(control). Statistical significance: * *p*-value < 0.05.

**Table 1 cells-11-01064-t001:** Clinico-morphological characteristics of EOC patients.

Sample	TNM	Grade	Stage (FIGO)	Histological Subtype
T1	T3cN0M1	High (G3)	IV	serous adenocarcinoma
T2	T3cN1M0	High (G3)	IIIC	serous papillary adenocarcinoma
T3	T3cN1M0	High (G3)	IIIC	serous papillary adenocarcinoma
T4	T1cN0M0	High (G2)	IC	serous adenocarcinoma
T5	T3cN0M1	High (G3)	IVA	serous papillary adenocarcinoma

**Table 2 cells-11-01064-t002:** Selected miRNAs differently expressed in EVs from the UA of EOC patients and healthy individuals. Criteria for selection: FC > 4, FDR-adjusted *p* < 0.05, logCPM > 9.

Gene ID	logFC	logCPM	F	*p* Value	FDR	Fold
hsa-miR-451a	−4.2758589	12.4477224	36.9140415	1.29 × 10^−9^	3.42 × 10^−6^	0.052
hsa-miR-542-3p	−3.6525866	9.5709767	23.3039465	0.0000014	0.001709212	0.080
hsa-miR-1246	3.8004008	9.0879369	22.5883317	0.0000024	0.001709212	13.933
hsa-miR-375-3p	3.2136512	11.7583475	22.0561752	0.0000027	0.001709212	9.277
hsa-miR-125a-5p	2.9175311	13.6969615	21.0651975	0.0000045	0.001709212	7.556
hsa-miR-320b	2.9883874	12.9093771	20.9288553	0.0000048	0.001709212	7.936
hsa-miR-200c-3p	2.9497286	13.1576673	20.7891165	0.0000052	0.001709212	7.726
hsa-miR-424-5p	−3.2575984	10.0135020	20.1278828	0.0000073	0.002145565	0.105
hsa-miR-200b-5p	3.3101791	9.7988733	18.3239500	0.0000187	0.004512432	9.919
hsa-miR-183-5p	3.0544179	9.9486138	17.8818248	0.0000238	0.005116818	8.308
hsa-miR-200b-3p	2.5691068	14.7872804	17.7661256	0.0000251	0.005116818	5.934
hsa-miR-320c	2.8136239	11.4890222	17.2009005	0.0000338	0.005592499	7.030
hsa-miR-224-5p	2.9448739	10.2072139	17.1046125	0.0000358	0.005592499	7.700
hsa-miR-450b-5p	−3.0021235	9.3416476	16.4403970	0.0000508	0.006731828	0.125
hsa-miR-320a-3p	3.0742866	9.1541706	16.2224191	0.0000565	0.006813182	8.423
hsa-let-7b-5p	2.3190987	14.6406996	14.6399033	0.0001311	0.012859111	4.990
hsa-miR-125b-5p	2.3615198	13.7812621	14.4991659	0.0001406	0.012859111	5.139
hsa-miR-182-5p	2.5967790	10.9627996	14.4123220	0.0001484	0.01311557	6.049
hsa-miR-143-3p	−2.1783923	15.8799851	13.8648943	0.0001978	0.01604251	0.221
hsa-miR-196b-5p	−2.4547336	11.9317925	13.8508279	0.0001996	0.01604251	0.182
hsa-miR-449c-5p	−2.7414632	9.7370987	13.1831605	0.0002831	0.020700409	0.150
hsa-miR-9-5p	2.5062128	10.4029623	13.0367742	0.0003083	0.020962645	5.681
hsa-miR-19b-3p	−2.6217121	9.0047643	12.4768255	0.0004129	0.026072094	0.162
hsa-miR-411-5p	−2.7085572	9.3188322	12.2746304	0.0004601	0.028203095	0.153
hsa-miR-10a-5p	2.0485626	14.9248514	11.8254458	0.0005854	0.033030914	4.137
hsa-miR-141-3p	1.9032256	9.1714047	10.9735792	0.0067931	0.040527411	3.740
hsa-miR-429	2.0748839	9.8455066	10.4926311	0.0006894	0.041667954	4.213
hsa-miR-23b-3p	−2.0119478	14.1412192	10.3749023	0.0009300	0.046277078	0.248
hsa-miR-199a-3p	−2.0198928	14.8878751	9.9660032	0.0011188	0.049553691	0.247

LogFC—logarithm of the relative change in miRNA expression, logCPM—logarithm of the number of miRNA reads per million, F—Fisher’s statistical significance test value, *p*-value—initial significance level value with probability *p*, FDR—*p*-value corrected (using Benjamini–Hochberg method) to account for false detection results, FC (fold change)—recalculated change in miRNA expression in folds. An increase (FC > 1, highlighted in red) and a decrease (FC < 1, highlighted in green) in the miRNA levels in EOC patients compared to healthy individuals are shown.

## Data Availability

The data presented in this study are available in the article and Appendix A.

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
