# Peer review of "Extracellular Vesicles from Uterine Aspirates Represent a Promising Source for Screening Markers of Gynecologic Cancers"

_cells, 2022, doi:10.3390/cells11071064_

Round 1
Reviewer 1 Report
I carefully read the article “Extracellular vesicles from uterine aspirates represent a promising source for screening markers of gynecologic cancers approval”.
First of all, the focus of the study is very current and important because ovarian cancer is still the leading cause of cancer deaths in women worldwide and moreover a screening method for early diagnosis is missing.
The study is well argued and below I want to highlight three points:
- the description of the process is very detailed (isolation of EVs, RNA extraction and sequencing, etc),
- the statistical analyses underneath the results give strength to the study,
- I confirm what you said at line 464 because this is the only article in the current literature which relates EVs to a gynecological cancer.
However, few clarifications could be useful.
- There are some bibliographical references missing:
- at line 49-55,
- at line 81-82, which evidences? Add references,
- at line 390 and 396,
- at line 504.
- What is the cost of the procedure? Is it suitable to become a screening method? Or is it too expensive and so it can be used only in women with risk factors? This could be a future topic to face.
- It's necessary to give more details about the category of patients involved;
- Maybe it could be helpful to provide some more information on the differences EVs found between samples of healthy subjects compared to the findings of patients with EOC;
Author Response
Dear Reviewer, we have carefully read your comments and have modified the text accordingly. Our responses are in italics.
However, few clarifications could be useful.
- There are some bibliographical references missing:
- at line 81-82, which evidences? Add references,
References added, reference numbers are [10–13]. Line numbers are still 81-82
- at line 390 and 396,
References added, reference numbers are [18] and [19]. Updated line numbers are 392 and 400
- at line 504.
References added, reference number is [56]. Updated line number is 493
- What is the cost of the procedure? Is it suitable to become a screening method? Or is it too expensive and so it can be used only in women with risk factors? This could be a future topic to face.
The data presented here are from a pilot study. We plan to continue the study using NGS small-RNA seq on an expanded sample of uterine aspirates from patients with ovarian cancer and a control group. We then hope to identify a panel of candidate miRNAs by RT-qPCR-related techniques using independent sampling. Based on the results of this step, we hope to select a set of miRNAs which can be further used as a test system for diagnosis of ovarian cancer by RT-qPCR.
The cost of the procedure itself for analysis of such a set of miRNAs will be relatively low, but it is important to realize that EV isolating is a rather complicated step (not so much expensive as requiring expensive equipment and time). Therefore, at first, such diagnosis can only be performed in specialized centers and certainly for women with risk factors. However, this approach may eventually find wider application if tested successfully.
- It's necessary to give more details about the category of patients involved;
We have added a table with the patients' clinicopathological data, including histological subtype, TNM, grade and stage of the disease, to the section "Materials and methods" - see Table 1 line 101
- Maybe it could be helpful to provide some more information on the differences EVs found between samples of healthy subjects compared to the findings of patients with EOC;
We found no statistically significant differences between EVs obtained from EOC patients and healthy donors. We added in the text that in addition to vesicle size, also no differences were observed in other characteristics such as EVs' concentration - line 275 and composition of exosomal markers - lines 300-301. This is an expected result since, as we stressed in the manuscript, there are currently no criteria that can be used to distinguish EVs of tumor origin.
In addition to the above corrections in line with your comments, we have made some additional technical corrections to the text (brackets, more correct representation of fractional numbers (in Fig.1 and in the text) and some others - see lines 250-254, 272-273.
Reviewer 2 Report
In this manuscript, Skryabin et al. investigated the potential of using uterine aspirates-derived EVs to search diagnostic markers for gynecological cancers. A number of miRNAs including those associated with ovarian cancer has been found differentially expressed in a pilot set of EOC and control samples. The topic is important and interesting. I do not have any major concerns, only some minor suggestions:
1. Could the authors specify more clearly whether/why your discovery is related to early diagnosis of gynecologic cancer?
2. I understand the focus of this research is about the comparison of EV miRNAs content. To be more systematic, is there any exosomal proteins/lipids discovered differentially expressed in EOC and control samples?
3. Please simplify the discussion part to make it more concise.
Author Response
Dear Reviewer, we have carefully read your comments and have modified the text accordingly. Our responses are in italics.
- Could the authors specify more clearly whether/why your discovery is related to early diagnosis of gynecologic cancer?
Early diagnosis of OC is one of the most important tasks of gynecologic oncology. From our point of view, in order to identify markers for early diagnosis of EOC, the first step is to identify EOC-associated molecules (in particular, exosomal microRNAs) as well as to confirm their significance for the pathogenesis of this disease. Our study is a pilot research, performed on 10 samples, which we did not divide according to the stage of the disease. The main finding of the work is the first evidence that EVs corresponding to exosomes can be isolated from uterine aspirates and that their molecular composition (in particular, the composition of microRNAs) differs significantly between samples from OC patients and healthy subjects. Next, we are going to conduct a study on an expanded sample from patients with different stages of the disease. As a result of this analysis, we plan to identify a panel of cancer-associated miRNAs and further confirm their involvement in the pathogenesis of EOC using experimental models. Among the miRNAs identified, we further plan to identify those whose levels are altered in the early stages and which can be used as a marker for early diagnosis of EOC.
To make this issue clearer, we have added several items:
- “The search for markers for early diagnosis of OC is therefore an extremely urgent task. The first evidence is emerging that exosomal miRNAs can serve as such markers.”- lines 477-478
- “Further studies using an expanded sample of uterine aspirates obtained from patients at different stages of the disease will help to identify exosomal microRNAs as potential markers of ovarian cancer, including for early diagnosis of EOC.” - lines 561-563
- I understand the focus of this research is about the comparison of EV miRNAs content. To be more systematic, is there any exosomal proteins/lipids discovered differentially expressed in EOC and control samples?
We did not investigate the composition of proteins and lipids in EVs from uterine aspirates, although both of these tasks seem very interesting, including in terms of searching for markers for the diagnosis of gynecological cancers. Regarding proteins, we investigated only the levels of exosomal markers in the EVs in order to confirm the nature of the EVs we were dealing with. Theoretically, it cannot be excluded that the spectrum of such proteins as well as their ratio in EVs isolated from aspirates of OC patients and healthy individuals could be different. We have observed similar changes, as well as differences in EV size, in particular for EVs isolated from the gastric juice of patients with gastric cancer (data to be published). However, no such differences were observed in this case. Consistent with your questions, we have pointed out the lack of such differences in the text of the manuscript - see lines 300-301.
- Please simplify the discussion part to make it more concise
We have significantly revised the Discussion section, making it clearer and more structured. We have also removed unnecessary details and simplified certain phrases. See lines 398-407, 410-416, 420-425, 435-443, 489-497, 517-533, 539-549.-We hope that this part has become more concise and better understood by the reader.
In addition to the above corrections in line with your comments, we have made some additional technical corrections to the text (brackets, more correct representation of fractional numbers (in Fig.1 and in the text) and some others – see lines 250-254, 272-273.
